# Inchworm-like Soft Robot with Multi-Responsive Bilayer Films

**DOI:** 10.3390/biomimetics8050443

**Published:** 2023-09-21

**Authors:** Xufeng Wang, Wei Pu, Ruichen Zhang, Fanan Wei

**Affiliations:** School of Mechanical Engineering and Automation, Fuzhou University, Fuzhou 350108, China; wangxufeng0610@163.com (X.W.); 1414390382@163.com (W.P.); zrc_10@163.com (R.Z.)

**Keywords:** biomimetic, soft robot, electric-moisture drive, GO-CNT/PE film, inchworm

## Abstract

As an important branch of robotics, soft robots have the advantages of strong flexibility, a simple structure, and high safety. These characteristics enable soft robots to be widely used in various fields such as biomedicine, military reconnaissance, and micro space exploration. However, contemporary soft crawling robots still face problems such as the single drive mode and complex external equipment. In this study, we propose an innovative design of an inchworm-like soft crawling robot utilizing the synergistic interaction of electricity and moisture for its hybrid dual-drive locomotion. The legs of the soft robot are mainly made of GO-CNT/PE composite film, which can convert its own volume expansion into a corresponding bending motion after being stimulated by electricity or moisture. Unlike other drive methods, it requires less power and precision from external devices. The combination of the two driving methods greatly improves the environmental adaptability of the soft robot, and we developed visible light as the driving method on the basis of the dual drive. Finally, we also verified the robot’s excellent load capacity, climbing ability, and optical drive effect, which laid the foundation for the application of soft robots in the future.

## 1. Introduction

In the past decade, there have been significant advancements in the field of soft robotics. The morphology and motion of natural organisms, utilizing flexible materials and external stimuli for power, have provided ample inspiration for soft robots. Unlike traditional robots that incorporate complex electronic components, power batteries, or precision computer chips, soft robots can be wirelessly controlled and miniaturized [1,2,3,4]. Inchworms are commonly found in nature and their motion pattern is among the simplest biological motion patterns, making it both straightforward and achievable to imitate their simple and easy motion.

Soft robots can be categorized into two types of actuation: tethered and untethered [5]. Tethered actuation involves the continuous supply of energy to the soft robot through external cables, which can include gas actuation [6], electroactive polymer actuation, and fluid actuation [7,8]. On the other hand, untethered actuation provides energy to the robot without the need for physical contact, relying on alternative energy sources such as light [9], chemical energy [10], magnetic fields [11], and moisture [12]. Tethered actuation presents several challenges, including bulky external equipment, complex control, complicated preparation, limited driving range [13], poor environmental adaptability, and unfriendliness [14], when compared to other actuation methods. In contrast, electricity and moisture offer various advantages as driving sources for soft robots, including easy acquisition, low cost, lack of pollution, and easy control.

Currently, electrically-driven soft robots that imitate inchworms primarily employ shape memory alloys (SMA) [15,16,17] and dielectric elastomers (DE) [18,19]. SMA-based soft robots consist of multiple metallic compounds [20,21]. These compounds undergo deformation and revert to their original martensitic structure upon exposure to heat or mechanical stimuli, which gives them the ability to remember their shape [15]. For example, Nguyen et al. developed a lightweight crawling robot called SMALLBug, weighing only 30 milligrams [16]. This robot, measuring 13 mm in length, incorporates an SMA wire and a carbon fiber sheet in its actuator. When driven at a high frequency of 20 Hz, it achieves an average velocity of 17 mm/s, equivalent to 1.3 times its body length per second. This agile robot demonstrates fast movement, providing valuable insights for the advancement of highly efficient robotic systems. Furthermore, Sascha et al. designed a DE-based crawling robot reinforced with additional fabric [19]. In experiments, this robot achieves a relative elongation rate of up to 2.4%, generating a force of 0.29 N, and attaining a movement speed of 28 mm/min. This demonstration showcases the potential of utilizing DE in the complex geometries of soft robots. However, although these technologies have the capability to achieve significant deformations and rapid motion, they are limited by their dependence on relatively fixed synthetic materials, which restricts the variety of driving mechanisms. Moreover, the need for higher driving voltages poses challenges to the advancements in miniaturization, safety, and micro-scale development.

Moisture-driven soft robots primarily employ crawling and rolling as their main modes of motion. Graphene oxide (GO) [22,23,24,25], MXene (Ti3C2Tx) [26], and other materials are commonly used for this purpose. GO stands out among traditional materials due to its unique structure and oxygen-containing functional groups [27]. These groups readily interact with water molecules, significantly enhancing GO’s hydrophilicity and facilitating quick moisture absorption or desorption processes [28]. Zhou et al. utilized 3D printing to create a GO gel for moisture-driven actuators [25]. In this method, the GO gel contracts only under constraints, ensuring structural uniformity. By inducing sharp variations in the alignment of the corners of three-dimensional constraints, gradient pores naturally form along the thickness direction, simplifying the fabrication of moisture-sensitive GO materials. The experimental results demonstrate that soft robots fabricated using these actuators achieve an average speed of 8.6 mm/min under moisture stimulation. Additionally, Yang et al. developed a composite film composed of dopamine-modified MXene and bacterial cellulose nanofibers, creating a structure resembling pearls and layers. This composite film exhibits high conductivity, exceptional tensile strength and toughness, water sensitivity, and rapid responsiveness [26]. They fabricated this film into a cylindrical rolling robot capable of carrying cargo while rolling at a relative humidity (RH) of 40%. This work establishes the foundation for the design and manufacturing of intelligent actuators with high conductivity, superior mechanical performance, and powerful actuation capabilities. However, current moisture-driven soft robots face several challenges, including the need for materials with high responsiveness, ensuring the stability and robustness of composite films, achieving uniform structures, and addressing limitations in the stability of moisture-driven systems.

To address these issues, we developed a hybrid film actuator consisting of CNT-GO and PE film. The hybrid film exhibits electrical, moisture, and light response capabilities due to the excellent electrical and visible light responsiveness of CNT and the moisture responsiveness of GO. The actuator demonstrates remarkable electrical and moisture reversible actuation, exhibiting large deformations and fast response capabilities, as well as some light actuation effects. By incorporating the hybrid film with PET film, we manufactured an electric and moisture-responsive soft robot inspired by inchworms, showcasing its motion capabilities and highlighting the potential of multi-drive films in bionic devices. This development presents new avenues for the advancement of multi-drive soft robots.

## 2. Materials and Methods

Figure 1 depicts the detailed fabrication process of the inchworm-like soft robot. To create the legs of the robot, a mixture of GO powder and CNT powder was prepared at a mass ratio of 1:1.5. Generally, the mass of GO is 0.0225 g, and 15 mL of deionized water was added to this mixture, forming a GO-CNT solution. The solution was then subjected to ultrasonic homogenization (JY92-IIN; Lichen Technology, Shaoxing, China) at a power of 35% for 8 min, with a cycle of 4 s on and 2 s off. Simultaneously, a PE tape measuring 50 mm in width and 100 μm in thickness was affixed to a horizontal acrylic base, ensuring a seamless connection between the PE film and the acrylic board. The 3D-printed mold with rectangular internal grooves (80 × 45 × 5 mm^3^ in size) was adhered to the PE film using adhesive. Afterwards, a volume of 15 mL of homogenized GO-CNT solution was poured into the mold. Subsequently, the mold was placed on a homogenizer to evenly distribute the solution throughout the mold at a speed of 60–100 r/min. Once the solution was evenly distributed, it was dried in a drying oven at 55 °C for 40 min, followed by an additional 60 min at 40 °C. Afterward, the film was naturally dried, removed from the mold, and cut into a “U” shape, with the two legs forming the input and output interfaces for voltage. As for the body of the robot, a rectangular shape measuring 30 × 25 mm^2^ was cut out from 0.1 mm thick PET film using a laser cutter (Muse Desktop CO_2_ Laser; Full Spectrum Laser, Las Vegas, NV, USA). The connection between the body and legs was established using double-sided conductive adhesive tape measuring 25 × 10 mm^2^. Additionally, a 0.05 mm copper wire, serving as the power source for the robot, was directly attached to the tape. Finally, the legs and body were assembled, and any excess tape was removed.

Following the aforementioned fabrication process, a multi-responsive bilayer film inchworm-like soft robot can be obtained. It should be noted that due to the pre-curved angles of the leg films, which exhibit minimal variation in bending angles, there is no need for additional bending procedures on the legs. Simply installing the pre-cut “U”-shaped legs onto the robot’s torso is sufficient.

## 3. Results and Discussion

The film exhibits a bending behavior with the stimuli. When electricity is applied, the current passes through the GO-CNT layer, resulting in heating due to the Joule effect. Heat is then transferred to the PE layer, causing it to subsequently heat up. Due to the different thermal expansion coefficients of the GO-CNT and PE layers, the volume change in the PE layer is greater, leading to asymmetric expansion between the two layers. This asymmetry causes the film to bend towards the GO-CNT layer since it experiences greater expansion. On the other hand, when moisture is introduced, the GO-CNT layer is capable of absorbing it while the PE layer shows no response. As a result, the GO-CNT layer absorbs moisture and expands asymmetrically, causing the film to bend towards the PE layer. Figure 2c illustrates the bending pattern of the film.

It is important to note that during the fabrication process, residual stress generates an initial bending angle *θ*_0_ in the composite film. As the applied voltage increases, the double-layer film exhibits a larger bending angle. However, the maximum bending angle *θ*_max_ is limited to 180° due to experimental constraints, as shown in Figure 2a,b. For the evaluation of the robot’s bending angles, we recorded the robot’s movements from a fixed position and then analyzed the motion data from the recorded videos using Kinovea software. It is worth noting that in Kinovea, we established a consistent motion coordinate system to ensure the accuracy of angle measurements.

### 3.1. Influence of Different Factors on Bending Performance of Composite Film

#### 3.1.1. The Impact of Mass Ratio of GO and CNT

The performance of the composite film is significantly affected by the mass ratio of GO and CNT. When the GO:CNT ratio is less than 1:1, the film tends to develop numerous small holes due to the low CNT content, resulting in poor conductivity and the loss of its electrical response ability. When the GO:CNT ratio is less than 1:2, an excessive CNT content leads to poor film-forming properties of the GO-CNT mixed solution, causing significant cracks on the film surface and rendering it impossible to produce. Accordingly, we conducted tests with five different GO:CNT mass ratios while maintaining a voltage of 12 V to evaluate the impact of varying ratios on the bending performance of the film. As shown in Figure 3a, a GO:CNT ratio of 1:1 yields the slowest response speed and the smallest *θ*_max_ due to the low CNT content, resulting in weak conductivity and an inadequate Joule heating effect. Thus, the bending angle is minimal, and the speed is comparatively slower. With increased CNT content, the film exhibits higher conductivity, improved Joule heating ability, and an enhanced electrical response. At a GO:CNT ratio of 1:2, the film demonstrates the fastest response speed, achieving *θ*_max_ within 5 s. Upon reaching the maximum bending angle, the power can be uniformly turned off after 35 s, and the film will naturally be restored to its initial state. For GO:CNT ratios exceeding 1:1, the recovery speed of the film remains relatively constant as the CNT content increases, suggesting a limited influence on the film’s recovery speed.

Figure 3b illustrates the impact of different GO:CNT mass ratios on the recovery performance of the film. After reaching *θ*_max_ at 35 s, the power was turned off, and 90% RH was introduced. Enhanced by moisture stimulation, the film’s recovery speed accelerated significantly, causing it to bend to a smaller angle than the initial position. Notably, the film with a GO:CNT ratio of 1:1 exhibits the smallest bending angle as its CNT content within the film is the lowest, resulting in the most significant change in angle. Other mass ratios of the film show similar recovery curves, but overall, the lower the proportion of CNT in the composite film, the smaller the minimum bending angle achievable under moisture stimulation, and the greater the range of bending angle change.

#### 3.1.2. The Impact of GO-CNT and PE Film Thickness

The GO-CNT/PE double-layer composite film consists of a GO-CNT layer and PE film, with the thickness of each layer significantly affecting the film’s performance. To ensure a PE film thickness of 100 μm and a GO:CNT mass ratio of 1:1.5, GO-CNT layers were produced with thicknesses of both 30 μm and 50 μm. Under a voltage of 12 V, the bending speed of the composite film with a 50 μm thick GO-CNT layer was slower compared to the film with a 30 μm thick layer, and it failed to reach the *θ*_max_, as shown in Figure 4a. This can be attributed to an increase in film thickness which results in increased strength but decreased toughness, leading to greater resistance to deformation and consequently reducing the maximum bending angle and speed at the same voltage.

Furthermore, the thickness of the PE film also has a crucial impact on the film’s performance. To investigate this, experiments were conducted with three different thicknesses of the PE film, while maintaining a GO-CNT film thickness of 30 μm. As depicted in Figure 4b, at 12 V voltage, the double-layer film exhibited the fastest response and recovery speed when the PE film thickness was 50 μm, whereas the film with a thickness of 100 μm showed the slowest response, with the bending speed not differing significantly. Additionally, the bending angle of the double-layer film was compared under varying thicknesses of the PE film after changing the voltage. From Figure 4c, it can be observed that even at a low voltage of 10 V, the 50 μm thick PE film was capable of reaching a bending angle of 124°. Moreover, at the same voltage, this film exhibited the largest bending angle. This is due to the fact that, with the same amounts of GO and CNT, a thinner PE film requires less energy to bend, making the composite film more susceptible to bending.

Although the composite films made of PE films with thicknesses of 50 μm and 70 μm demonstrate a high response speed, they suffer from insufficient film strength, resulting in susceptibility to deformation and twisting when subjected to multiple cycles of voltage and moisture stimulation. This leads to a decline, and even damage, in the composite film’s performance. Furthermore, the GO-CNT/PE composite film produced with a 100 μm thick PE film exhibits a slower response speed. However, its strength is adequate to meet the basic motion requirements of the robot. To assess the stability of the GO-CNT/PE composite film comprising a 100 μm thick PE film, a series of bending performance tests was conducted under 12 V voltage and moisture stimulation, as depicted in Figure 4d. The experimental findings reveal that even after numerous cycles of stimulation, the robot’s maximum bending angle remains at 180°, with the minimum bending angle under moisture stimulation around 15°. There is no apparent decrease in bending performance, thus demonstrating excellent fatigue resistance and an extremely long service life for the composite film with a 100 μm thick PE film. Moreover, the GO-CNT/PE film exhibits exceptional stability. Even after being left undisturbed for a month, the maximum bending angle of the film remains at 180° during the bending performance test.

### 3.2. Continuous Periodic Motion of Inchworm-like Robot and the Effects of Voltage and Moisture on Its Motion

The inchworm-like robot was positioned on the experimental platform and subjected to horizontal motion through the external stimulation of 18 V voltage and 90% RH. The contact points A, located between the front of the robot’s body and the ground, as well as the contact points B, existing between the legs and the ground, along with the angle α formed between the body and the ground, were employed as markers or marked angles for trajectory tracking. Figure 5a offers a visual representation of these markers and marked angles. As depicted in Figure 5b, it is evident that upon the initial application of voltage, the tensile force acting on point A exceeds the frictional force, resulting in its movement in the −X direction. Similarly, with point B, the tension surpasses the frictional force, gradually propelling it in the +X direction. Subsequently, the voltage input ceases, and moisture is introduced. Consequently, point A initiates movement in the +X direction, while point B either remains stationary or moves in the −X direction. When point A assumes a stationary state, the moisture stimulation is discontinued, leading to the return of point B to a forward motion and attainment of the equilibrium position. Consequently, the robot reverts to its original state, completing one cycle of motion.

Figure 5c,d illustrate the relationship between the robot’s body length and angle α, which display clearly recognizable periodic variations. This outcome signifies that the soft robot demonstrates relatively stable movement in each cycle and is capable of achieving continuous and stable motion. Figure 5e visually illustrates the soft robot’s motion under 18 V voltage and 90% RH. Evidently, the body and legs of the soft robot exhibit noticeable deformation during movement, with one cycle of robot motion lasting approximately 30 s.

Voltage plays a primary role in influencing the motion of the soft robot. The bending of its legs propels the robot’s movement when subjected to voltage stimulation. Figure 6a illustrates that a higher voltage leads to the increased bending speed of the legs. Additionally, under the same moisture stimulation, a higher voltage results in a shorter cycle of robot motion. Consequently, within a given timeframe, the soft robot covers a greater distance and achieves a higher average speed. Furthermore, with higher voltage, the legs are capable of bending forward over a longer distance during the initial phase of each cycle, allowing the robot to travel a greater distance within each cycle. Figure 6b clearly demonstrates that increased external voltage correlates with a higher average speed of the robot, with a maximum average speed reaching 0.219 mm/s, and its motion can be seen in Appendix A. However, excessive input voltage should be avoided as it can generate excessive heat, potentially damaging the soft robot and impairing its ability to move.

The impact of moisture on the soft robot is similar to that of voltage, specifically accelerating the legs’ recovery speed. Figure 6c,d reveal that a higher RH leads to greater horizontal displacement per unit time, resulting in an overall faster average speed. The notable distinction is that, as moisture increases, the increment in average speed becomes smaller, which contrasts with the effect of voltage.

### 3.3. Multidimensional Capability Testing of Inchworm-like Robot

The study of inchworm-like robots aims to achieve various applications, such as environmental pollutant degradation or cell and drug transportation within the human body [29,30,31]. Hence, we also examined the cargo transportation capability of the inchworm-like robot. The “cargo” carried by the robot comprised polydimethylsiloxane (PDMS) in a 10:1 ratio of liquid A to B. Once solidified, rectangles with a length and width of 1 cm were cut from the cargo, as depicted in Figure 7a, weighing approximately 0.032 g on average. The robot’s body material is PET film, and the viscosity of PDMS allows it to be placed on the body without slipping, eliminating the need for an extra cargo storage device, as shown in Figure 7b. The experimental results revealed that the robot, at a voltage of 21 V and 90% RH, can carry up to five cargos with a total weight of approximately 0.16 g. Furthermore, the weight of the cargo is around 0.6 times its own weight. Figure 7c demonstrates the relationship between displacement and time. Notably, the robot experiences significantly reduced movement speed when loaded. At maximum load, the average movement speed of the soft robot is 0.076 mm/s, roughly half the speed it achieves when unloaded, as presented in Figure 7d.

Climbing poses a formidable challenge for the robot during actual motion; therefore, its climbing ability necessitates testing. To evaluate this ability, a ladder-shaped ramp was designed as a right-angled triangle with a height of 1 mm and length of 5 mm, as illustrated in Figure 8a. Placing the ladder on a slope allows the robot to crawl upward from the bottom. The soft robot exhibited a maximum climbing angle of 70° during testing, as shown in Figure 8b. At this angle, its climbing trajectory is depicted in Figure 8c, and it managed to ascend two steps in 33 s.

The leg film of the inchworm-like robot consists of GO-CNT/PE film, where CNT offers exceptional heat exchange and light absorption properties, thereby enabling the realization of light-driven soft robots. Consequently, we also evaluated its light-driving ability. A xenon lamp served as the light source, and the light intensity was adjustable by modifying the output current. Figure 8d illustrates the positive correlation between the light source illumination intensity and the output current. By determining the current and light power density, we selected five levels of light power density for testing. As Figure 8e showcases, the robot’s average speed gradually increases with the strengthening of the light power density. This occurs because higher light power densities result in greater heat energy absorption and conversion per unit time by the CNT, facilitating larger and faster bending of the leg film and consequently leading to swifter movement speeds, with a maximum speed of 0.181 mm/s.

## 4. Conclusions

In order to broaden the application range and control method of soft crawling robots, we adopted a simple method to fabricate GO-CNT/PE composite films and designed an inchworm-like robot based on an electro-wetting composite actuation strategy. The robot uses volume expansion triggered by electrical and moisture stimuli to achieve locomotion. The contribution of this study is that it greatly reduces the need for external equipment when driving soft robots, realizes the movement mode of electric and moisture hybrid dual drives, and discusses the possibility and effect of optical drives. In this study, the main factors affecting the bending deformation of the composite film and the movement speed of the robot were tested, and the relevant laws and relationships were summarized. The cargo transportation ability and climbing ability exhibited by the inchworm-like robot provide a basis for the application of multi-power hybrid drive soft robots in the future.

## Figures and Tables

**Figure 1 biomimetics-08-00443-f001:**
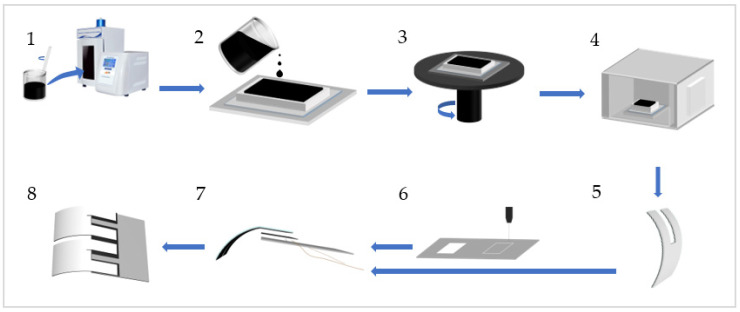
The fabrication process of the inchworm-like robot. (**1**) Place a GO-CNT mixture solution with a mass ratio of 1:1.5 into an ultrasonic homogenizer for uniform dispersion; (**2**) pour the uniformly dispersed solution into a mold, with an acrylic board at the bottom, a PE film in the middle, and the mold on top; (**3**) position the solution-filled mold on a spin coater for processing, ensuring even distribution of the solution within the mold to form a uniformly thick film; (**4**) place the spun-coated film together with the mold into a drying oven for drying; (**5**) retrieve the dried film and cut it into the desired shapes as needed; (**6**) utilize a laser cutting machine to cut PET film for creating the robot’s torso; (**7**) assemble the torso and legs using connectors; and (**8**) obtain the mimic millipede soft robot.

**Figure 2 biomimetics-08-00443-f002:**
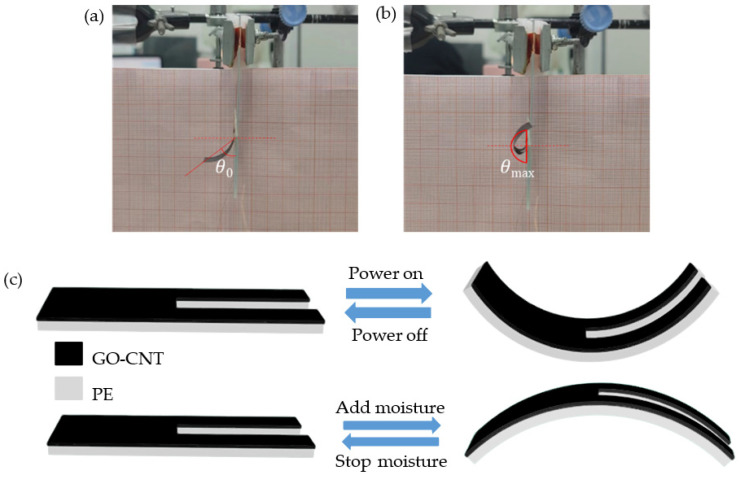
GO-CNT/PE double-layer composite film experimental device. (**a**) Composite film bending initial angle diagram; (**b**) composite film bending maximum angle diagram; and (**c**) deformation mechanism of composite films under the action of electricity and moisture.

**Figure 3 biomimetics-08-00443-f003:**
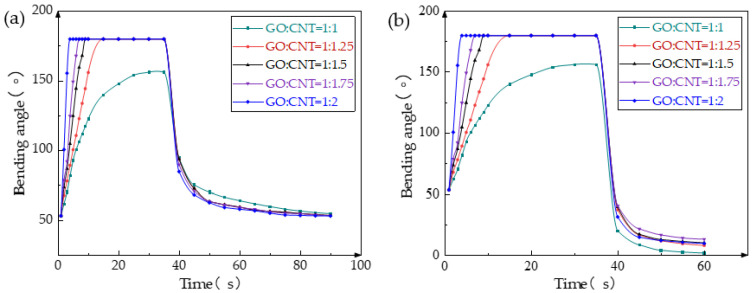
Response curves of films with different GO-CNT mass ratios under 12 V voltage. (**a**) Response curves of films with different GO-CNT mass ratios under voltage stimulation and natural recovery; and (**b**) response curves of films with different GO-CNT mass ratios under voltage stimulation and 90% RH recovery.

**Figure 4 biomimetics-08-00443-f004:**
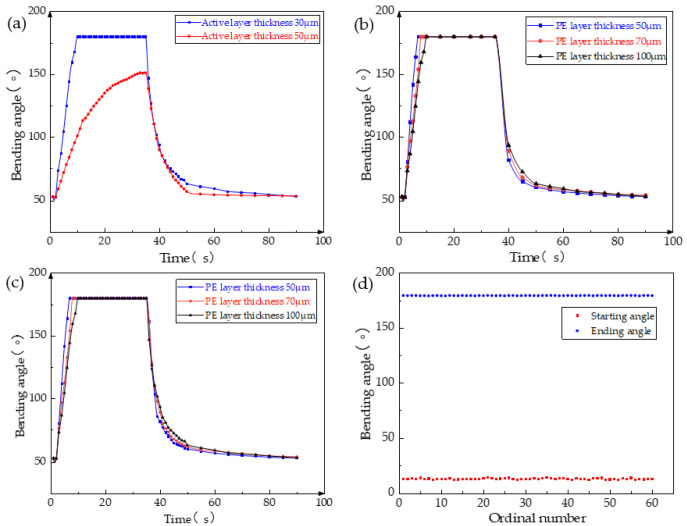
Test results of different thicknesses of active layer and passive layer of GO-CNT/PE bilayer composite film. (**a**) Effect of passive layer thickness on the response speed of the composite film; (**b**) effect of passive layer thickness on the response speed of the composite film; (**c**) effect of passive layer thickness on the bending angle of the composite film at different voltages; and (**d**) composite film cycle experiments under voltage and moisture.

**Figure 5 biomimetics-08-00443-f005:**
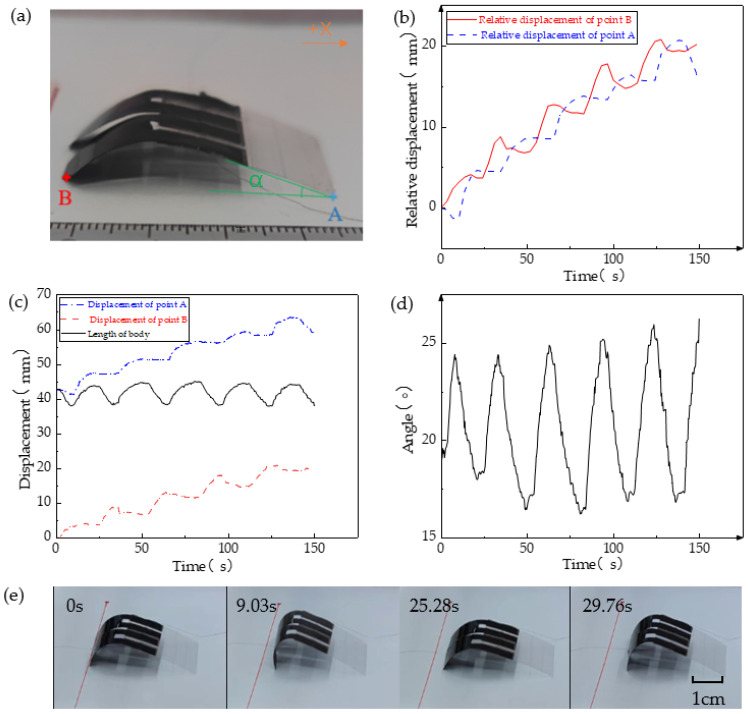
Cyclical motion of the mimic inchworm soft robot. (**a**) Explanation of tracking points A, B, angle α, and direction of motion; (**b**) A and B relative displacement locus; (**c**) A, B point displacement and the relationship between body length and time; (**d**) the relationship between the angle α between the robot body and the ground and time; and (**e**) a screenshot of a cycle motion of a mimic inchworm soft robot.

**Figure 6 biomimetics-08-00443-f006:**
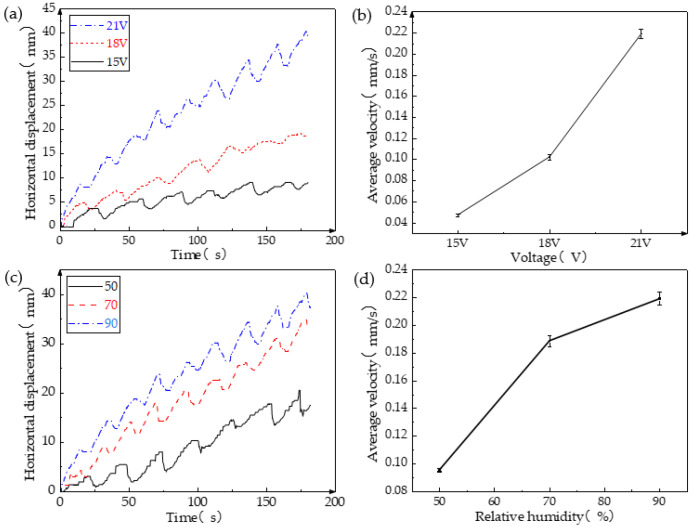
Influence of voltage and moisture on periodic motion of robot. (**a**) The relationship between the displacement and time of the soft crawling robot under different voltages; (**b**) the relationship between the average speed and time of the soft crawling robot under different voltages; (**c**) the relationship between the displacement and time of the soft crawling robot under different RH; and (**d**) the relationship between the average speed of the soft crawling robot and RH.

**Figure 7 biomimetics-08-00443-f007:**
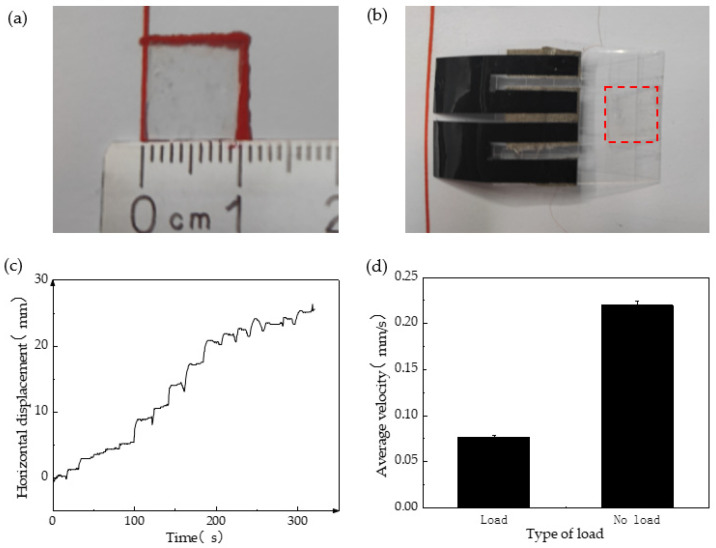
Cargo transfer test of soft crawling robot. (**a**) The cargo cube made of PDMS; (**b**) the cargo is placed in the area selected by the red square; (**c**) the relationship between the displacement and time of the cargo during transportation; and (**d**) the comparison of the average speed of the soft crawling robot with and without load.

**Figure 8 biomimetics-08-00443-f008:**
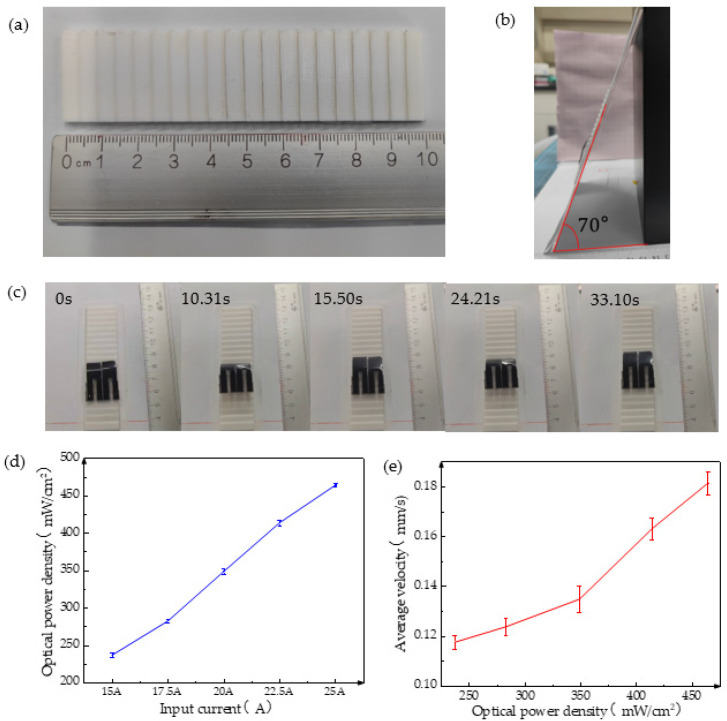
Climbing performance and light-driven effect of soft crawling robot. (**a**) Dimensional drawing of the 3D-printed ladder; (**b**) maximum climbing angle of the soft crawling robot; (**c**) diagram of the climbing motion process of the soft crawling robot; (**d**) relationship between output current and optical power density; and (**e**) relationship between optical power density and average speed of soft crawling robot.

## Data Availability

Not applicable.

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
