# Peer review of "Inchworm-like Soft Robot with Multi-Responsive Bilayer Films"

_biomimetics, 2023, doi:10.3390/biomimetics8050443_

Round 1

Reviewer 1 Report

This study proposes an innovative design of an inchworm-like soft crawling robot utilizing the synergistic interaction of electricity and moisture for its hybrid dual-drive locomotion. I have few comments for the authors:

1. How do the authors measure the bending angle?

2. Please demonstrate the mechanism deign drawing in the paper.

Author Response

Reply to: “1.      How do the authors measure the bending angle?

Thanks for the constructive suggestion from the reviewer. We have revised our conclusion to make it more insightful and more enlightening. The conclusion part is changed from:

“It is important to note that during the fabrication process, residual stress generates an initial bending angle ?0 in the composite film. As the applied voltage increases, the double-layer film exhibits a larger bending angle. However, the maximum bending angle ?max is limited to 180° due to experimental constraints, as shown in Figure 2(a) and 2(b).”

To

“It is important to note that during the fabrication process, residual stress generates an initial bending angle ?0 in the composite film. As the applied voltage increases, the dou-ble-layer film exhibits a larger bending angle. However, the maximum bending angle ?max is limited to 180° due to experimental constraints, as shown in Figure 2(a) and 2(b). For the evaluation of the robot's bending angles, we recorded the robot's movements from a fixed position and then analyzed the motion data from the recorded videos using Kinovea software. It's worth noting that in Kinovea, we established a consistent motion coordinate system to ensure the accuracy of angle measurements.”

Reply to: “2.Please demonstrate the mechanism deign drawing in the paper.

Thanks for the reviewer’s helpful suggestion. The basic principle of electricity and moisture deforming composite films is as follows. When electricity is applied, the current passes through the GO-CNT layer, resulting in heating due to the Joule effect. Heat is then transferred to the PE layer, causing it to heat up subsequently. Due to the different thermal expansion coefficients of the GO-CNT and PE layers, the volume change in the PE layer is greater, leading to asymmetric expansion between the two layers. This asymmetry causes the film to bend towards the GO-CNT layer since it experiences greater expansion. On the other hand, when humidity is introduced, the GO-CNT layer is capable of absorbing it while the PE layer shows no response. As a result, the GO-CNT layer absorbs humidity and expands asymmetrically, causing the film to bend towards the PE layer. we have added the relevant deformation mechanism diagram to Figure 2(c). and the updated figure is displayed as below.

Besides, the figure caption is revised as following.

Figure 2. GO-CNT/PE double-layer composite film experimental device. (a) Composite film bending initial angle diagram; (b) Composite film bending maximum angle diagram; (c) Deformation mechanism of composite films under the action of electricity and moisture.

The conclusion part is changed from:

“On the other hand, when moisture is introduced, the GO-CNT layer is capable of absorbing it while the PE layer shows no response. As a result, the GO-CNT layer absorbs moisture and expands asymmetrically, causing the film to bend towards the PE layer. Figure 2(a) illustrates the bending pattern of the film.”

To

“On the other hand, when moisture is introduced, the GO-CNT layer is capable of absorbing it while the PE layer shows no response. As a result, the GO-CNT layer absorbs moisture and expands asymmetrically, causing the film to bend towards the PE layer. Figure 2(c) illustrates the bending pattern of the film.”

Reviewer 2 Report

1.     In “Materials and Methods” Section, only the ratios of GO and CNT are mentioned; while type and quantity of the solution are also crucial for the final experiment results. So as to accurately obtain the GO-CNT mixture solution, the authors are highly suggested to supplement the information as described above.

2.     In the section of "Materials and Methods," since there is only one subsection, there is no need for the subheading "2.1 Fabrication of Bilayer Film Actuators and Soft Robot."

3.     "Humidity" primarily refers to the moisture content in the atmosphere, while "Moisture" specifically denotes the moisture present in the air. In this manuscript, the authors seem to utilize the moisture in the air for propulsion. Therefore, please pay attention to the precise usage of these two terms.

4.     "To investigate this, experiments were conducted with three different thicknesses of the PE film (50 μm, 70 μm, and 100 μm), while maintaining a GO-CNT film thickness of 30 μm."The three different thicknesses of the PE film can be observed in the subsequent figures, so there is no need to present this information in the text.

5.     In the electric actuation experiments, the applied voltage went as high as 21 V. Then, please provide the corresponding power when the voltage is 21 V.

Minor revision

Author Response

Reply to: “1. In “Materials and Methods” Section, only the ratios of GO and CNT are mentioned; while type and quantity of the solution are also crucial for the final experiment results. So as to accurately obtain the GO-CNT mixture solution, the authors are highly suggested to supplement the information as described above.

Thanks to the reviewer for the valuable advice. The solvent we use is deionized water. Generally, the mass of GO is 0.0225g and the volume of deionized water is 15ml. The conclusion part is changed from:

“Figure 1 depicts the detailed fabrication process of the Inchworm-like soft robot. To create the legs of the robot, a mixture of GO powder and CNT powder was prepared at a mass ratio of 1:1.5. Deionized water was added to this mixture, forming a GO-CNT solution.”

To

“Figure 1 depicts the detailed fabrication process of the Inchworm-like soft robot. To create the legs of the robot, a mixture of GO powder and CNT powder was prepared at a mass ratio of 1:1.5. Generally, the mass of GO is 0.0225g, and 15ml of deionized water was added to this mixture, forming a GO-CNT solution.”

Reply to: “2. In the section of "Materials and Methods," since there is only one subsection, there is no need for the subheading "2.1 Fabrication of Bilayer Film Actuators and Soft Robot.”

Thank you for your suggestion, the relevant content has been modified. The conclusion part is changed from:

“2. Materials and Methods

2.1 Fabrication of Bilayer Film Actuators and Soft Robot

Figure 1 depicts the detailed fabrication process of the Inchworm-like soft robot. To create the legs of the robot, a mixture of GO powder and CNT powder was prepared at a mass ratio of 1:1.5. Deionized water was added to this mixture, forming a GO-CNT solution. The solution was then subjected to ultrasonic homogenization (JY92-IIN; Lichen Technology, China) at a power of 35% for 8 minutes, with a cycle of 4 seconds on and 2 seconds off. Simultaneously, a PE tape measuring 50 mm in width and 100 μm in thickness was affixed to a horizontal acrylic base, ensuring a seamless connection between the PE film and the acrylic board. The 3D-printed mold with rectangular internal grooves (80 × 45 × 5 mm³ in size) is adhered to the PE film using adhesive. Afterwards, a volume of 15 ml of homogenized GO-CNT solution is poured into the mold. Subsequently, the mold was placed on a homogenizer to evenly distribute the solution throughout the mold at a speed of 60-100 r/min. Once the solution was evenly distributed, it was dried in a drying oven at 55 ℃ for 40 minutes, followed by an additional 60 minutes at 40 ℃. Afterward, the film was naturally dried, removed from the mold, and cut into a "U" shape, with the two legs forming the input and output interfaces for voltage. As for the body of the robot, a rectangular shape measuring 30 × 25 mm² was cut out from a 0.1 mm thick PET film using a laser cutter(MUSE; FSL3D, USA). The connection between the body and legs was established using a double-sided conductive adhesive tape measuring 25 × 10 mm². Additionally, a 0.05 mm copper wire, serving as the power source for the robot, was directly attached to the tape. Finally, the legs and body were assembled, and any excess tape was removed. ”

To

“2. Materials and Methods

Figure 1 depicts the detailed fabrication process of the Inchworm-like soft robot. To create the legs of the robot, a mixture of GO powder and CNT powder was prepared at a mass ratio of 1:1.5. Deionized water was added to this mixture, forming a GO-CNT solution. The solution was then subjected to ultrasonic homogenization (JY92-IIN; Lichen Technology, China) at a power of 35% for 8 minutes, with a cycle of 4 seconds on and 2 seconds off. Simultaneously, a PE tape measuring 50 mm in width and 100 μm in thickness was affixed to a horizontal acrylic base, ensuring a seamless connection between the PE film and the acrylic board. The 3D-printed mold with rectangular internal grooves (80 × 45 × 5 mm³ in size) is adhered to the PE film using adhesive. Afterwards, a volume of 15 ml of homogenized GO-CNT solution is poured into the mold. Subsequently, the mold was placed on a homogenizer to evenly distribute the solution throughout the mold at a speed of 60-100 r/min. Once the solution was evenly distributed, it was dried in a drying oven at 55 ℃ for 40 minutes, followed by an additional 60 minutes at 40 ℃. Afterward, the film was naturally dried, removed from the mold, and cut into a "U" shape, with the two legs forming the input and output interfaces for voltage. As for the body of the robot, a rectangular shape measuring 30 × 25 mm² was cut out from a 0.1 mm thick PET film using a laser cutter(MUSE; FSL3D, USA). The connection between the body and legs was established using a double-sided conductive adhesive tape measuring 25 × 10 mm². Additionally, a 0.05 mm copper wire, serving as the power source for the robot, was directly attached to the tape. Finally, the legs and body were assembled, and any excess tape was removed. ”

Reply to: “3. "Humidity" primarily refers to the moisture content in the atmosphere, while "Moisture" specifically denotes the moisture present in the air. In this manuscript, the authors seem to utilize the moisture in the air for propulsion. Therefore, please pay attention to the precise usage of these two terms.”

We totally agree with what the reviewer pointed out. We have a general description of the term “relative humidity” but also “humidity” replacement “Moisture”.

Reply to: “4. To investigate this, experiments were conducted with three different thicknesses of the PE film (50 μm, 70 μm, and 100 μm), while maintaining a GO-CNT film thickness of 30 μm."The three different thicknesses of the PE film can be observed in the subsequent figures, so there is no need to present this information in the text.”

We greatly apologized for having used a confusing expression. The conclusion part is changed from:

“To investigate this, experiments were conducted with three different thicknesses of the PE film (50 μm, 70 μm, and 100 μm), while maintaining a GO-CNT film thickness of 30 μm."The three different thicknesses of the PE film can be observed in the subsequent figures, so there is no need to present this information in the text.”

To

“To investigate this, experiments were conducted with three different thicknesses of the PE film, while maintaining a GO-CNT film thickness of 30 μm."The three different thicknesses of the PE film can be observed in the subsequent figures, so there is no need to present this information in the text.”

Reply to: “5. In the electric actuation experiments, the applied voltage went as high as 21 V. Then, please provide the corresponding power when the voltage is 21 V.”

Firstly, thanks for the insightful question from the reviewer. When the voltage is 21V, the measured current is 0.04A, therefore, its power is 0.84W. However, I think this is not a necessary parameter to display, so it has not been modified in the article.

Reviewer 3 Report

In this manuscript, inspired by the inchworm, the authors proposed and realized a actuator driven by both humidity and electricity. This work conducted sustainable actuation experiments for the actuator. The results demonstrated the nice locomotion capacity of the soft actuator. In general, this work is interesting. But, before acceptance for publication, the authors need to answer my concern as listed below.

1.     The soft crawling robot fabricated seems to be extremely soft. Then, what the largest loading capacity of the robot? And, the authors are encouraged to discuss the possible contradiction between the bending performance of the actuator and the loading capacity.

2.     As shown in Figure 6a and Figure 8d, the actuation voltage and current are very large. Would this level of voltage and current potentially damage the robot?

3.     There are many errors in the references. For example, in Ref 15, 16 and 17, “Ieee” should be “IEEE”. Please go through the whole part, and correct errors, if any.

4.     There are in fact three actuation approaches, i.e., electricity, moisture and light. Why opt for a hybrid drive combining electricity and moisture, rather than other combinations like electricity-light or light-moisture ?

5. In the abstract, "Unlike other drive methods, it requires less power and precision from external devices."  what is less precision ?

no

Author Response

Reply to: “1. The soft crawling robot fabricated seems to be extremely soft. Then, what the largest loading capacity of the robot? And, the authors are encouraged to discuss the possible contradiction between the bending performance of the actuator and the loading capacity.”

Thanks for the insightful suggestion from the reviewer. As mentioned in the article, after experiments, the robot can load up to five pieces of goods when the voltage is 21 V, RH is 90%, and its total weight is about 0.16 g, which is equivalent to 0.6 of its own weight. As the load increases, the bending performance continues to decrease.

Reply to: “2.As shown in Figure 6a and Figure 8d, the actuation voltage and current are very large. Would this level of voltage and current potentially damage the robot?.”

Thanks for the great question from the reviewer. The voltage in Figure 6a is the voltage directly provided to the soft robot. Under this high voltage, the machine is moving in a relatively overloaded state and can complete related movements, but the working time cannot be too long. The current in Figure 8d refers to the current provided to the xenon lamp, not the current directly provided to the robot. The light intensity generated by this level of current will not cause damage to the robot.

Reply to: “3. There are many errors in the references. For example, in Ref 15, 16 and 17, “Ieee” should be “IEEE”. Please go through the whole part, and correct errors, if any..”

Greatly thank the reviewer for the recommendation of those references. After a careful reading of the references, we found them valuable to our article and added them into the manuscript. The citation positions are listed as follows:

“4.Trivedi, D.; Lotfi, A.; Rahn, C. Geometrically Exact Models for Soft Robotic Manipulators. IEEE.T.Robot 2008, 24, 773-780.[CrossRef]

  1. Bena, R.M.; Nguyen, X.T.; Calderón, A.A.; Rigo, A.; Pérez-Arancibia, N.O. SMARTI: A 60-mg Steerable Robot Driven by High-Frequency Shape-Memory Alloy Actuation. IEEE.Robot.Autom.Let 2021, 6, 8173-8180.[CrossRef]
  2. Nguyen, X.T.; Calderón, A.A.; Rigo, A.; Ge, J.Z.; Pérez-Arancibia, N.O. SMALLBug: A 30-mg Crawling Robot Driven by a High-Frequency Flexible SMA Microactuator. IEEE.Robot.Autom.Let 2020, 5, 6796-6803.[CrossRef]
  3. Singh, P.; Ananthasuresh, G.K. A Compact and Compliant External Pipe-Crawling Robot. IEEE.T.Robot 2013, 29, 251-260.[CrossRef]”

Reply to: “4. There are in fact three actuation approaches, i.e., electricity, moisture and light. Why opt for a hybrid drive combining electricity and moisture, rather than other combinations like electricity-light or light-moisture ?”

Firstly, we greatly appreciate for the reviewer’s insightful criticism. Because electricity and light both convert energy into thermal energy, causing the double-layer film to bend asymmetrically. Since the film is rectangular, expansion mainly occurs in the length direction, so the film will bend toward the GO-CNT layer. Under moisture stimulation, the GO-CNT layer will expand due to the hygroscopic effect of GO, while the PE layer will not respond to moisture stimulation. In this case, the GO-CNT layer will increase in volume due to expansion, causing asymmetric expansion and causing the film to bend toward the PE layer. Therefore, it can only be a combination of electricity-humidity or light-humidity driving. Because the effect of electric driving is better than that of light driving, the electric-humidity driving combination was chosen. The relevant content has been clearly stated in the article.

Reply to: “5. In the abstract, "Unlike other drive methods, it requires less power and precision from external devices."  what is less precision ?

Thanks for the useful criticism from the reviewer. Taking magnetically controlled robots as an example, most magnetically controlled robots need to move under the control of a three-dimensional magnetic field, and the three-dimensional magnetic field is generated by three Helmholtz coils. When the Helmholtz coil magnetic field generation system generates a larger magnetic field, the required power and manufacturing accuracy are also greatly increased. Compared with the external equipment required by these drive methods, our electric-moisture drive only requires a DC power supply that can provide stable low voltage and a relatively stable moisture generator.
